# $\Delta S^z = 2$ quantum magnetization discontinuities proportional in number to the spin s in $C_{60}$: Origin and the role of symmetry

Nikolaos P. Konstantinidis⋆

Department of Mathematics and Natural Sciences, The American University of Iraq, Sulaimani, Kirkuk Main Road, Sulaymaniyah, Kurdistan Region, Iraq

⋆ npknpk1111@gmail.com

## Abstract

The quantum antiferromagnetic Heisenberg model on the fullerene $C_{60}$ in a magnetic field has $4s$ ground-state magnetization discontinuities with $\Delta S^z = 2$ as a function of the spin quantum number $s$ that disappear at the classical limit. The molecule can be seen as the fullerene $C_{20}$ with interpentagon interactions that generate the discontinuities when sufficiently strong. The discontinuities originate from the antiferromagnetic Ising limit for both molecules. The results show how spatial symmetry dictates the magnetic response of the $I_h$ fullerene molecules.

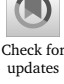

## 1  Introduction

Fullerenes are allotropes of carbon whose most representative member is $C_{60}$ [1–9]. $C_{60}$ superconducts when doped with alkali metals [10]. Electron correlations are important for doped

$C_{60}$, and they are accurately described within the framework of the Hubbard model (see [11] and references therein). According to estimates for the on-site repulsion $U$, $C_{60}$ belongs to the intermediate-$U$ regime of the Hubbard model [12–15], but its large $U$-limit, the Heisenberg model, is expected to qualitatively capture the spin correlations [16].

In the quantum antiferromagnetic Heisenberg model the total ground-state magnetization along an external field axis $S^z$ changes discontinuously at specific field values typically by $\Delta S^z = 1$. In cases of spin-space anisotropy the energy is more efficiently minimized along certain directions, and this can lead to discontinuities with $\Delta S^z > 1$. It is of particular interest when such magnetization discontinuities occur in the absence of magnetic anisotropy. These are solely due to the frustrated connectivity of the magnetic interactions, directly reflecting the topology of the structure hosting the magnetic units.

The antiferromagnetic Heisenberg model has been extensively used to model the magnetic properties of low-dimensional frustrated topologies [17–22]. In fullerene molecules frustration is introduced by the twelve pentagons that each molecule has, with the number of hexagons increasing linearly with size [1]. The most symmetric fullerene molecules have icosahedral $I_h$ symmetry like $C_{60}$. These are minimally frustrated among the fullerenes if one excludes the Platonic solid dodecahedron [23–26] that only has pentagons [27,28], and have also been found to share the ground-state magnetic response. At the classical level there are two magnetization discontinuities in an external field when all exchange interactions are equal, with the exception of the dodecahedron that has three [16,29,30]. In the extreme quantum limit where the individual spin magnitude $s = \frac{1}{2}$ and 1, the dodecahedron has respectively one and two discontinuities with $\Delta S^z = 2$ [29]. A high-field ground-state magnetization jump with $\Delta S^z = 2$ was established to be a common feature of the $I_h$ fullerenes for $s = \frac{1}{2}$ when all exchange interactions are equal [30]. For $I_h$-fullerenes bigger than the dodecahedron it is not possible to calculate the magnetic response for not-so-high $S^z$ due to computational limitations imposed by the size of the Hilbert space. Relatively small fullerene molecules with different symmetry have only pronounced magnetization plateaus when $s = \frac{1}{2}$ [31,32]. Magnetization discontinuities with $\Delta S^z > 1$ have also been observed in the case of extended systems [33–40].

The dual of the dodecahedron, the icosahedron, has also been shown to possess a magnetization discontinuity at the classical lowest-energy configuration in an external field, which disappears at the extreme quantum limit [41]. This discontinuity can be understood from a structural point of view, as the icosahedron can be viewed as a closed strip of a triangular lattice with two additional spins attached [42]. Such a structural explanation of the magnetization discontinuities is not obvious for the dodecahedron. It can neither apply to the discontinuities of fullerene molecules where all pentagons are located at their ends and their body has the form of a nanotube, comprising only of hexagons [11].

To investigate the origin of the $\Delta S^z = 2$ magnetization discontinuities of the dodecahedron for $s = \frac{1}{2}$ and 1 the quantum anisotropic Heisenberg model (AHM) is considered. Its antiferromagnetic isotropic limit is approached from the corresponding Ising limit by gradually increasing the strength of the interactions in the $xy$ plane. Karľová *et al.* calculated the ground-state magnetization response of the $s = \frac{1}{2}$ AHM on the dodecahedron for different values of the anisotropy [43], extending the calculation done at the isotropic limit [29]. In the present paper it is shown that there is a strong ground-state magnetization discontinuity at the antiferromagnetic Ising limit at the saturation field. This discontinuity is very closely monitored as the fluctuations in the $xy$ plane are switched on. Degenerate perturbation theory on the $xy$-plane interactions shows that the saturation-field jump develops into the $\Delta S^z = 2$ high-field magnetization discontinuity five spin-flips away from saturation for any finite $s$, with the discontinuity disappearing at the classical limit $s \to \infty$, demonstrating its pure quantum nature. Then evolving continuously away from the antiferromagnetic Ising limit by increasing the strength of the $xy$ coupling shows that the jump survives beyond the isotropic Heisenberg

limit for $s = \frac{1}{2}$ and 1. For increasingly higher $s$ the $xy$-plane fluctuations are strong enough to confine the discontinuity closer and closer to the Ising limit.

The calculation is then extended to the next-bigger $I_h$ fullerene, the truncated icosahedron $C_{60}$, which has also been found to support the $\Delta S^z = 2$ ground-state magnetization discontinuity five spin flips away from saturation for $s = \frac{1}{2}$ and uniform interactions [30]. Here the relative strength of the two symmetrically independent interactions is allowed to vary and relates to how strongly neighboring pentagons are coupled, since they are isolated from one another and do not share any vertices or edges, as in the dodecahedron. Degenerate perturbation theory shows that the $\Delta S^z = 2$ discontinuity is again generated infinitesimally away from the antiferromagnetic Ising limit for arbitrary $s$ if the interpentagon coupling is at least equal to $\tan^{-1}(0.01143\pi)$ times the intrapentagon one. However, its existence at the antiferromagnetic isotropic limit requires the two couplings to be roughly equal in order for the $\Delta S^z = 2$ discontinuity to survive for $s = \frac{1}{2}$. Increasing $s$ again works against the discontinuity, but making the interpentagon coupling strong enough allows the jump to appear for arbitrary $s$, with the required minimum interpentagon coupling increasing with $s$ and the AHM approaching closer and closer its strong interpentagon-coupling or dimer limit. This demonstrates that $C_{60}$ can behave like its smaller symmetric relative, $C_{20}$, if the interpentagon interactions are strong enough, and shows the close connection between symmetry and magnetic properties. Furthermore since the next-bigger $I_h$ fullerene, the chamfered dodecahedron [1], also has a $\Delta S^z = 2$ five-spin-flips away from saturation ground-state jump in a magnetic field for $s = \frac{1}{2}$ when its two symmetrically unique interactions are equal [30], it is expected that bigger molecules with $I_h$ symmetry will follow the discontinuity pattern of $C_{60}$. In their case the interpentagon coupling is mediated by exchange interactions of one or more kinds. The similarities in the ground-state magnetic response of the $I_h$ fullerene molecules demonstrate a strong correlation between symmetry and magnetic behavior for this family, which can allow the prediction of the properties of larger molecules, which are impossible to treat numerically, from the ones of their smaller relatives.

In order to investigate in more detail the behavior close to the dimer limit where the $\Delta S^z = 2$ ground-state discontinuity is favored and the interpentagon is much stronger than the intrapentagon interaction, degenerate perturbation theory on the latter is considered for arbitrary $s$. The Hilbert space of the AHM on $C_{60}$ is enormous, however degenerate perturbation theory allows the calculation of the lowest energies for the whole range of $S^z$ in this limit, as long as $s$ is not too big. $C_{60}$ then reduces to an icosidodecahedron of dimers interacting weakly with one another. For $s = \frac{1}{2}$ it is found that apart from the five-spin-flips away from saturation $\Delta S^z = 2$ discontinuity at the isotropic AHM limit, another one exists at low fields with the $S^z = 5$ sector never including the ground state in a field. This is a manifestation of the singlet-triplet or hole-particle symmetry between low and high magnetic fields after the projection to the lowest-energy singlet and triplet states of the dimers has been made [44–46]. Degenerate perturbation theory for arbitrary $s$ shows that the number of $\Delta S^z = 2$ ground-state discontinuities grows as $4s$, with the pattern of lower and higher-field jumps repeating in the magnetization curve $2s$ times and disappearing at the classical limit. These discontinuities are expected to survive away from the dimer limit, as has been shown for the one close to saturation, and show how insight on the magnetic response for lower-$S^z$ sectors not directly accessible with diagonalization can be deduced by degenerate perturbation theory on the dimer limit. Again, since symmetry and strong interpentagon coupling are important for the appearance of the jumps, it is expected that the latter will be general features of the $I_h$-fullerene molecules.

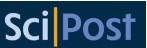

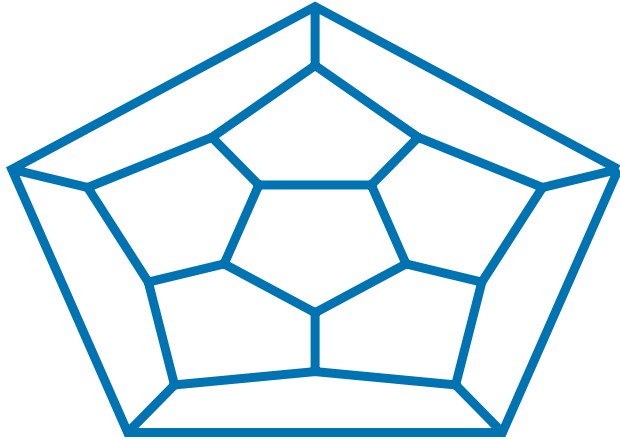

Figure 1: Planar projection of the dodecahedron $C_{20}$.

## 2 Model

The Hamiltonian of the quantum AHM in a magnetic field $\vec{h}$ with a single spin $\vec{s}_i$, $i = 1, \ldots, N$ located on each of the $N$ vertices of a molecule is

$$H = \sum_{\langle ij \rangle} J_{ij} \left[ \sin \omega \left( s_i^x s_j^x + s_i^y s_j^y \right) + \cos \omega s_i^z s_j^z \right] - h \sum_{i=1}^{N} s_i^z . \tag{1}$$

The symbol $\langle \; \rangle$ indicates that interactions are limited to nearest-neighbors and have strength equal to $J_{ij}$ for spins $i$ and $j$ connected by an edge of the molecule. The Ising coupling along the $z$ axis is scaled with $\cos\omega$ and the coupling in the $xy$ plane with $\sin\omega$, and $0 \leq \omega < \frac{3}{4}\pi$ is taken. The antiferromagnetic limits are the Ising for $\omega = 0$, the isotropic Heisenberg for $\omega = \frac{\pi}{4}$, and the XX for $\omega = \frac{\pi}{2}$. The interactions are taken to obey the molecular symmetry, with two edges connected by a symmetry operation of the $I_h$ point group corresponding to the same interaction strength. The magnetic field is directed along the $z$ axis. Hamiltonian (1) is block-diagonalized by taking into account $S^z$ and its spatial and spin symmetries, and the lowest-lying level in each sector is then found with Lanczos diagonalization [29]. At the Ising limit the lowest-energy states of Hamiltonian (1) in the different $S^z$ sectors are degenerate. It is also possible to perturb away from this limit with the interactions in the $xy$ plane by simultaneously taking the Hamiltonian symmetries into account, and calculate the first-order energy correction with degenerate perturbation theory [47, 48].

## 3 Dodecahedron $C_{20}$

The dodecahedron, whose planar projection is shown in Fig. 1, is a Platonic solid [23], and all its $N = 20$ vertices are geometrically equivalent. It consists of twelve pentagons, and is the smallest fullerene in the form of $C_{20}$ [1, 24–26]. All of its edges are symmetrically equivalent, making all bonds $J_{ij}$ in Hamiltonian (1) equal, and it is taken $J_{ij} \equiv 1$ from now on. At the antiferromagnetic Ising limit $\omega = 0$ the lowest energy of Hamiltonian (1) belongs to both the $S^z = 0$ and $4s$ subsectors, making the lowest-lying levels from $S^z = 1$ to $4s - 1$ excited (Tables 1, 2, and 3). The ground-state energy as a function of $S^z$ is linear, resulting in a strong magnetization jump $\Delta S^z = 16s$ to saturation (Figs 2 and 3).

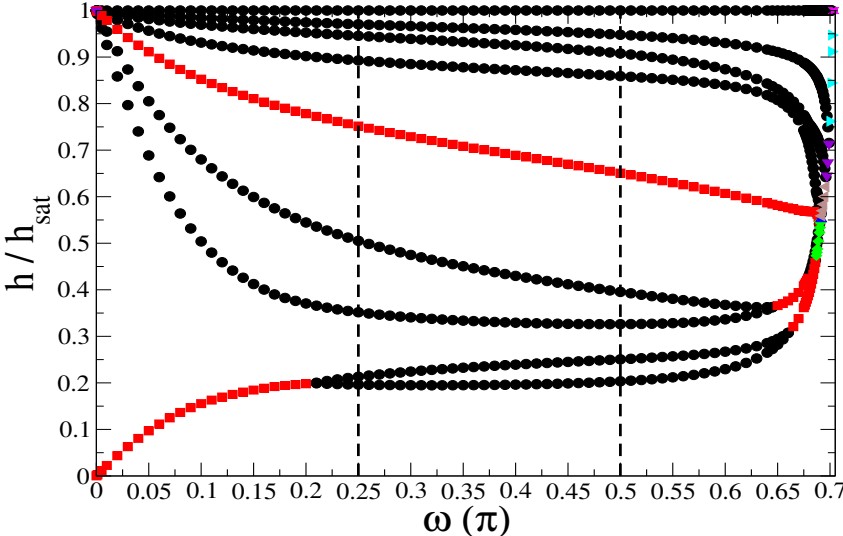

Figure 2: Magnetic fields $h$ over the saturation field $h_{sat}$ for which ground-state magnetization discontinuities occur as a function of $\omega$ in the ground state of Hamiltonian (1) for $C_{20}$ for $s = \frac{1}{2}$. The (black) circles correspond to discontinuities with $\Delta S^z = 1$, the (red) squares with $\Delta S^z = 2$, the (green) diamonds with $\Delta S^z = 3$, the (blue) triangles up with $\Delta S^z = 6$, the (brown) triangles left with $\Delta S^z = 7$, the (violet) triangles down with $\Delta S^z = 8$, the (cyan) triangles right with $\Delta S^z = 9$, and the (magenta) ×'s with $\Delta S^z = 10$. The (black) dashed lines show the isotropic Heisenberg ($\omega = \frac{\pi}{4}$) and the XX ($\omega = \frac{\pi}{2}$) limit.

Table 1: Hamiltonian (1) with $h = 0$ for $C_{20}$ for $s = \frac{1}{2}$. The columns list the sector $S^z$, the lowest energy $E_0$ at the Ising limit $\omega = 0$, its degeneracy (deg.), the first-order degenerate perturbation theory energy correction $\Delta E_1$ on $\omega$, its degeneracy (deg.), and the irreducible representation (irrep.) of the $I_h$ symmetry group it belongs. $\Delta E_1$ has been calculated with double-precision accuracy but less digits are shown.

| $S^z$ | $E_0$ | deg. | $\Delta E_1$ | deg. | irrep. |
|---|---|---|---|---|---|
| 0 | $-\frac{9}{2}$ | 240 | $-(\sqrt{3} + \frac{1}{2})$ | 1 | $A_u$ |
| 1 | -4 | 900 | -2.64146 | 3 | $T_{2u}$ |
| 2 | $-\frac{9}{2}$ | 5 | 0 | 5 | $A_g, F_g$ |
| 3 | -3 | 320 | -2.56533 | 4 | $F_u$ |
| 4 | $-\frac{3}{2}$ | 1240 | -3.68687 | 1 | $A_g$ |
| 5 | 0 | 1912 | -3.54419 | 4 | $F_g$ |
| 6 | $\frac{3}{2}$ | 1510 | -3.59294 | 1 | $A_u$ |
| 7 | 3 | 660 | -3.02938 | 3 | $T_{1u}$ |
| 8 | $\frac{9}{2}$ | 160 | -2.13276 | 5 | $H_g$ |
| 9 | 6 | 20 | $-\frac{\sqrt{5}}{2}$ | 3 | $T_{2u}$ |
| 10 | $\frac{15}{2}$ | 1 | 0 | 1 | $A_g$ |

Perturbing away from the Ising limit with the interactions in the $xy$ plane in first order ($\omega \to 0$) decreases the $S^z = 0$-energy but not the $S^z = 4s$ for $s = \frac{1}{2}$ (Table 1), resulting in a low-field $\Delta S^z = 4s$ magnetization jump (Fig. 2). For $s = 1$ and $\frac{3}{2}$ first-order perturbation theory does not resolve this degeneracy (Tables 2 and 3), but Lanczos diagonalization shows that

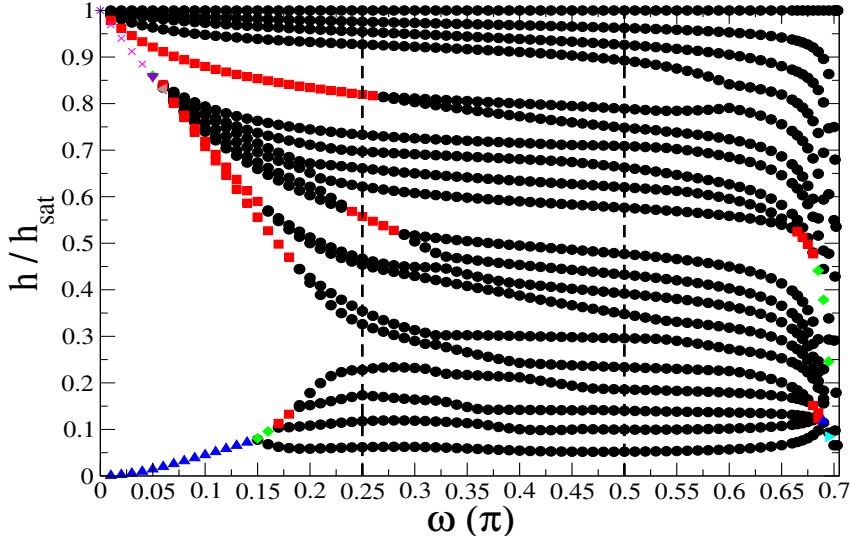

Figure 3: Magnetic fields $h$ over the saturation field $h_{sat}$ for which ground-state magnetization discontinuities occur as a function of $\omega$ in the ground state of Hamiltonian (1) for $C_{20}$ for $s = 1$. The (black) circles correspond to discontinuities with $\Delta S^z = 1$, the (red) squares with $\Delta S^z = 2$, the (green) diamonds with $\Delta S^z = 3$, the (blue) triangles up with $\Delta S^z = 4$, the (brown) triangles left with $\Delta S^z = 6$, the (violet) triangles down with $\Delta S^z = 7$, the (cyan) triangles right with $\Delta S^z = 9$, the (magenta) ×'s with $\Delta S^z = 10$, and the (indigo) stars with $\Delta S^z = 16$. The (black) dashed lines show the isotropic Heisenberg ($\omega = \frac{\pi}{4}$) and the XX ($\omega = \frac{\pi}{2}$) limit.

Table 2: Same as Tab. 1 for $s = 1$.

| $S^z$ | $E_0$ | deg. | $\Delta E_1$ | deg. | irrep. | $S^z$ | $E_0$ | deg. | $\Delta E_1$ | deg. | irrep. |
|---|---|---|---|---|---|---|---|---|---|---|---|
| 0 | -18 | 240 | 0 | 240 | all | 11 | 3 | 41120 | -4.20815 | 30 | all but $A_g, A_u$ |
| 1 | -17 | 1440 | -2 | 60 | all but $A_g$ | 12 | 6 | 41475 | -5.22625 | 10 | $A_g, F_g, H_g$ |
| 2 | -16 | 4080 | -3.81284 | 30 | all but $T_{1g}, T_{2g}, A_u$ | 13 | 9 | 32920 | -6.25465 | 20 | $A_g, F_g, H_g, T_{1u}, T_{2u}, F_u$ |
| 3 | -17 | 60 | -1 | 30 | all but $A_g, A_u$ | 14 | 12 | 20520 | -7.37374 | 1 | $A_g$ |
| 4 | -18 | 5 | 0 | 5 | $A_g, F_g$ | 15 | 15 | 9932 | -7.08838 | 4 | $F_g$ |
| 5 | -15 | 40 | 0 | 40 | all | 16 | 18 | 3650 | -7.18588 | 1 | $A_u$ |
| 6 | -12 | 460 | 0 | 460 | all | 17 | 21 | 980 | -6.05876 | 3 | $T_{1u}$ |
| 7 | -9 | 2520 | $-\sqrt{2}$ | 60 | all but $A_u$ | 18 | 24 | 180 | -4.26553 | 5 | $H_g$ |
| 8 | -6 | 8310 | $-\sqrt{5}$ | 120 | all | 19 | 27 | 20 | $-\sqrt{5}$ | 3 | $T_{2u}$ |
| 9 | -3 | 18920 | -3.12602 | 40 | all | 20 | 30 | 1 | 0 | 1 | $A_g$ |
| 10 | 0 | 31852 | -3.62611 | 120 | all | | | | | | |

the low-field $\Delta S^z = 4s$ discontinuity also occurs when $s = 1$ (Fig. 3). Magnetization jumps occur when the energy differences between the lowest-energy states of successive $S^z$ sectors do not increase with increasing $S^z$. Since the ground-state energy varies linearly with $S^z$ at the unperturbed limit, the energy differences between the lowest-energy levels in successive $S^z$ sectors are determined by the perturbative corrections. That first-order perturbation theory can not resolve the degeneracies away from the high-$S^z$ region is also demonstrated by the high degeneracy of the energy corrections in Tables 2 and 3, which belong to multiple irreducible

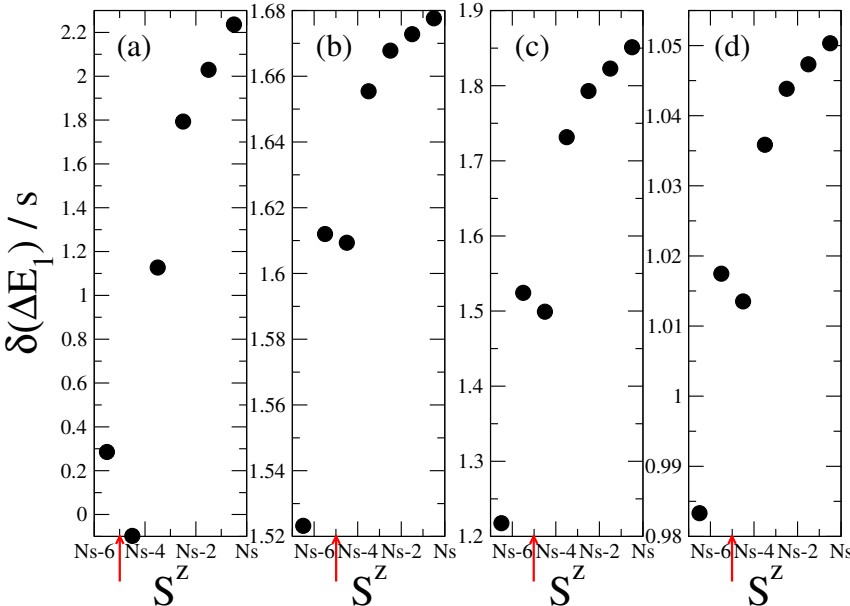

Figure 4: Differences of the first-order degenerate perturbation theory corrections of the energy over the spin magnitude $\frac{\delta(\Delta E_1)}{s} = \frac{\Delta E_1(S^z+1)}{s} - \frac{\Delta E_1(S^z)}{s}$ between successive $S^z$ sectors close to saturation for arbitrary $s$ for (a) $C_{20}$ (Table 4) and (b), (c), and (d) for $C_{60}$ for $\phi = \frac{\pi}{50}$, $\frac{\pi}{4}$, and $\frac{49}{100}\pi$ respectively (Table 6) away from the Ising limit $\omega = 0$ of Hamiltonian (1) for $h = 0$. The (red) solid arrows show the locations of the $\Delta S^z = 2$ magnetization discontinuities.

Table 3: Same as Tab. 1 for $s = \frac{3}{2}$.

| $S^z$ | $E_0$ | deg. | $\Delta E_1$ | deg. | irrep. | $S^z$ | $E_0$ | deg. | $\Delta E_1$ | deg. | irrep. |
|---|---|---|---|---|---|---|---|---|---|---|---|
| 0 | -40.5 | 240 | 0 | 240 | all | 16 | 4.5 | 250840 | -4.78375 | 30 | all but $T_{1g}, T_{2g}, A_u$ |
| 1 | -39 | 1440 | $-\frac{3}{2}$ | 720 | all | 17 | 9 | 301680 | -5.43916 | 120 | all |
| 2 | -37.5 | 4320 | -3 | 720 | all | 18 | 13.5 | 320160 | -6.24360 | 120 | all |
| 3 | -36 | 9980 | $-\frac{9}{2}$ | 340 | all | 19 | 18 | 300400 | -6.31222 | 30 | all but $A_g, A_u$ |
| 4 | -37.5 | 360 | -3 | 75 | all | 20 | 22.5 | 249032 | -7.83938 | 10 | $A_g, F_g, H_g$ |
| 5 | -39 | 60 | $-\frac{3}{2}$ | 30 | all but $A_g, A_u$ | 21 | 27 | 181860 | -7.83938 | 20 | $A_g, F_g, H_g$ $T_{1u}, T_{2u}, F_u$ |
| 6 | -40.5 | 5 | 0 | 5 | $A_g, F_g$ | 22 | 31.5 | 116315 | -9.38197 | 20 | $A_g, F_g, H_g$ $T_{1u}, T_{2u}, F_u$ |
| 7 | -36 | 40 | 0 | 40 | all | 23 | 36 | 64560 | -9.38197 | 20 | $A_g, F_g, H_g$ $T_{1u}, T_{2u}, F_u$ |
| 8 | -31.5 | 180 | 0 | 180 | all | 24 | 40.5 | 30680 | -11.06061 | 1 | $A_g$ |
| 9 | -27 | 880 | 0 | 880 | all | 25 | 45 | 12232 | -10.63257 | 4 | $F_g$ |
| 10 | -22.5 | 3570 | 0 | 3570 | all | 26 | 49.5 | 3970 | -10.77881 | 1 | $A_u$ |
| 11 | -18 | 11480 | $-\frac{3}{2}\sqrt{2}$ | 60 | all but $A_u$ | 27 | 54 | 1000 | -9.08814 | 3 | $T_{1u}$ |
| 12 | -13.5 | 29800 | $-\frac{3}{2}\sqrt{2}$ | 360 | all | 28 | 58.5 | 180 | -6.39829 | 5 | $H_g$ |
| 13 | -9 | 64040 | $-\frac{3}{2}\sqrt{5}$ | 120 | all | 29 | 63 | 20 | $-\frac{3}{2}\sqrt{5}$ | 3 | $T_{2u}$ |
| 14 | -4.5 | 116695 | $-\frac{3}{2}\sqrt{5}$ | 600 | all | 30 | 67.5 | 1 | 0 | 1 | $A_g$ |
| 15 | 0 | 183232 | -4.68903 | 40 | all | | | | | | |

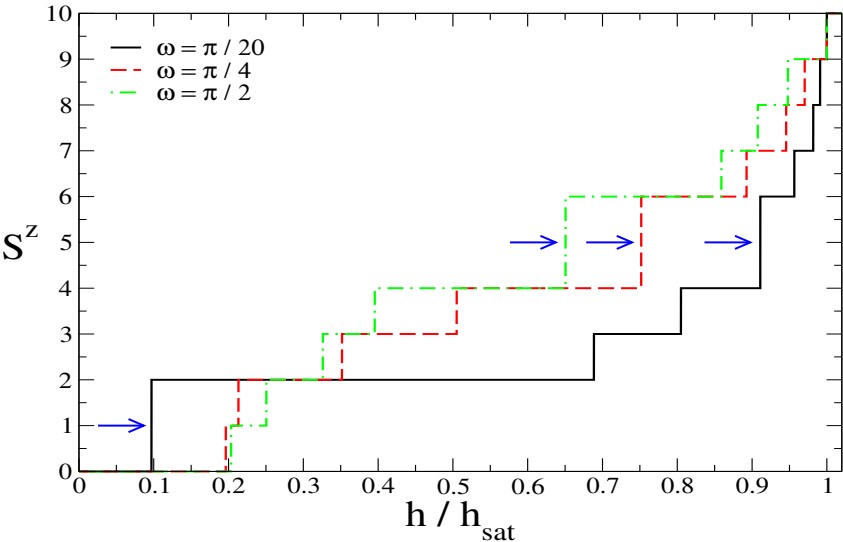

Figure 5: $S^z$ as a function of the magnetic field $h$ over the saturation field $h_{sat}$ in the ground state of Hamiltonian (1) for $C_{20}$ for $s = \frac{1}{2}$. The (black) solid line corresponds to $\omega = \frac{\pi}{20}$, the (red) long-dashed line to $\omega = \frac{\pi}{4}$, and the (green) long-dashed-dotted line to $\omega = \frac{\pi}{2}$. The (blue) solid arrows point to the discontinuities with $\Delta S^z = 2$.

representations [49]. First-order perturbation theory shows the existence of a magnetization jump between $4s$ and $Ns - 6$ for $s = 1$ and $\frac{3}{2}$. Fig. 3 shows that the jump survives for higher values of $\omega$ for $s = 1$.

In contrast to the perturbative results for lower $S^z$, from $S^z = Ns - 6$ to saturation the lowest-energy states belong to a single irreducible representation, which does not change with increasing $s$. The first-order perturbative corrections in these $S^z$ sectors are multiples of $s$ (Table 4), making the magnetization response close to saturation the same for any $s$. Calculating the energy differences between the high-field successive sectors shows that the $S^z = Ns - 5$ sector, the one with five spin-flips away from saturation, never becomes the ground state in the field, irrespective of $s$, resulting in a $\Delta S^z = 2$ discontinuity (Fig. 4(a)). The energy differences between the sectors scale with $s$, while the energies at the classical limit scale as $s^2$, showing that the discontinuity is a pure quantum effect which occurs for arbitrary finite $s$, but disappears at the classical limit $s \to \infty$. It is noted that for every $s = 1$-level with $4s < S^z < Ns - 6$ there is a corresponding $s = \frac{3}{2}$-level in the analogous $S^z$ range so that the ratio of their first-order perturbative energy corrections equals the ratio of their $s$ values.

The dodecahedron is small enough to calculate the lowest-energy state in each $S^z$-sector with Lanczos diagonalization for $s = \frac{1}{2}$ and $1$, allowing to monitor the evolution of all magnetization discontinuities away from first-order perturbation theory on the antiferromagnetic Ising limit. The magnetization response for $s = \frac{1}{2}$ and $1$ is shown in Figs 2 and 3, with the magnetization curves for $\omega = \frac{\pi}{20}$, $\frac{\pi}{4}$, and $\frac{\pi}{2}$ plotted in Figs 5 and 6. Karľová *et al.* have provided the plot for $s = \frac{1}{2}$ up to $\omega = \frac{\pi}{4}$ [43]. As the coupling in the $xy$ plane gets stronger the high-field discontinuity survives at the isotropic Heisenberg limit $\omega = \frac{\pi}{4}$ for both values of $s$. The other discontinuities disappear, with the exception of a $\Delta S^z = 2$ discontinuity around $S^z = 9$ for $s = 1$, that reenters a little before the isotropic limit and traces back to the $\Delta S^z = 10$ discontinuity of weak $\omega$.

Monitoring the $\Delta S^z = 2$ discontinuities away from the isotropic limit for higher $\omega$ shows that the two $s = 1$ jumps quickly disappear, while the $s = \frac{1}{2}$ discontinuity survives up to the XX limit where $\omega = \frac{\pi}{2}$ and the Ising interaction is zero. Further increasing $\omega$ makes the interaction

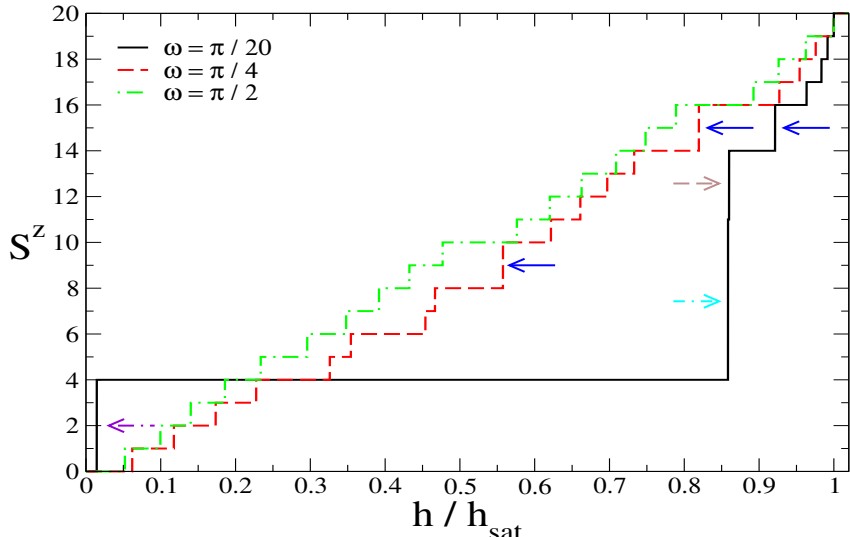

Figure 6: $S^z$ as a function of the magnetic field $h$ over the saturation field $h_{sat}$ in the ground state of Hamiltonian (1) for $C_{20}$ for $s = 1$. The (black) solid line corresponds to $\omega = \frac{\pi}{20}$, the (red) long-dashed line to $\omega = \frac{\pi}{4}$, and the (green) long-dashed-dotted line to $\omega = \frac{\pi}{2}$. The (blue) solid arrows point to the discontinuities with $\Delta S^z = 2$, the (brown) long-dashed arrow to the discontinuity with $\Delta S^z = 3$, the (violet) long-dashed-dotted arrow to the discontinuity with $\Delta S^z = 4$, and the (cyan) double-dashed-dotted arrow to the discontinuity with $\Delta S^z = 7$.

Table 4: Hamiltonian (1) with $h = 0$ for $C_{20}$ for arbitrary $s$. The columns list the sector $S^z$, the first-order degenerate perturbation theory energy correction per spin magnitude $\frac{\Delta E_1}{s}$ on $\omega$ (away from the Ising limit $\omega = 0$), its degeneracy (deg.), and the irreducible representation (irrep.) of the $I_h$ symmetry group it belongs. $\Delta E_1$ has been calculated with double-precision accuracy but less digits are shown.

| $S^z$ | $\frac{\Delta E_1}{s}$ | deg. | irrep. |
|---|---|---|---|
| $Ns-6$ | $-7.37374$ | 1 | $A_g$ |
| $Ns-5$ | $-7.08838$ | 4 | $F_g$ |
| $Ns-4$ | $-7.18588$ | 1 | $A_u$ |
| $Ns-3$ | $-6.05876$ | 3 | $T_{1u}$ |
| $Ns-2$ | $-4.26553$ | 5 | $H_g$ |
| $Ns-1$ | $-\sqrt{5}$ | 3 | $T_{2u}$ |
| $Ns$ | $0$ | 1 | $A_g$ |

along the $z$ axis ferromagnetic, and eventually the discontinuity disappears at $\omega = 0.68562\pi$. The detrimental effect of the $xy$-plane fluctuations on the $\Delta S^z = 2$ discontinuity away from the Ising limit is getting stronger with $s$ as shown in Fig. 7, which plots the highest $\omega$ value for which the jump survives as a function of $s$ (Table 5). The jump does not survive at the isotropic Heisenberg limit for $s > 1$.

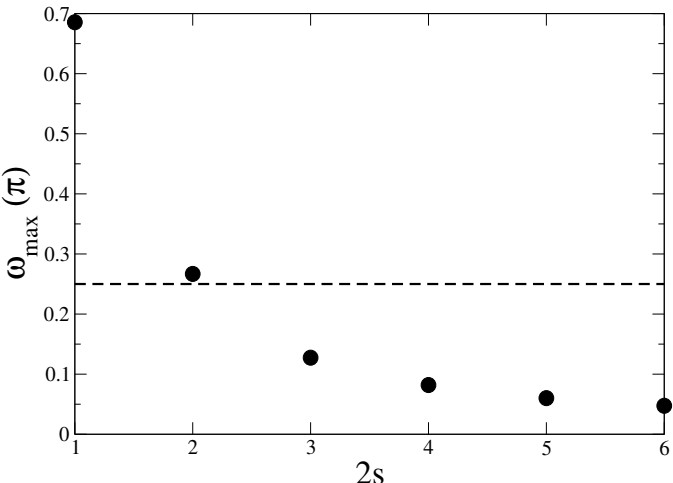

Figure 7: Maximum value $\omega_{max}$ as a function of $s$ for which the ground state of Hamiltonian (1) for $C_{20}$ has a $\Delta S^z = 2$ magnetization discontinuity five spin flips away from saturation. The dashed line shows the isotropic Heisenberg limit $\omega = \frac{\pi}{4}$.

Table 5: Maximum value $\omega_{max}$ for different $s$ for which the ground state of Hamiltonian (1) for $C_{20}$ has a $\Delta S^z = 2$ magnetization discontinuity between $S^z = Ns-6$ and $Ns-4$. The jump does not survive at the isotropic Heisenberg limit $\omega = \frac{\pi}{4}$ for $s > 1$.

| $s$ | $\omega_{max}(\pi)$ |
|---|---|
| $\frac{1}{2}$ | 0.68562 |
| 1 | 0.26685 |
| $\frac{3}{2}$ | 0.12745 |
| 2 | 0.08186 |
| $\frac{5}{2}$ | 0.06007 |
| 3 | 0.04739 |

## 4 Truncated Icosahedron $C_{60}$

The truncated icosahedron, whose planar projection is shown in Fig. 8, has $N = 60$ and is an Archimedean solid [50], having all vertices geometrically equivalent and two different types of polygons, pentagons and hexagons. It is the most representative fullerene in the form of $C_{60}$ [1–8]. It has two symmetrically unique types of edges, which correspond to two independent exchange interactions in Hamiltonian (1). The first type links vertices of the same pentagon and is taken to have strength $J_1 \equiv cos\phi$ (blue thick lines in Fig. 8), while the second vertices that belong to different pentagons and has strength $J_2 \equiv sin\phi$ (red thin lines in Fig. 8) with $0 \leq \phi \leq \frac{\pi}{2}$, interpolating between isolated pentagons and isolated dimers. Unlike the dodecahedron the pentagons do not share vertices but rather interact via the interpentagon bonds, which represent the interaction strength between neighboring pentagons.

Very close to the isolated pentagon limit $\phi$ is small and first-order degenerate perturbation theory away from the Ising limit shows that there is no $\Delta S^z = 2$ ground-state discontinuity five spin flips away from saturation. A minimum value of $\phi$ is required for the discontinuity to appear within first-order perturbation theory, and it exists for $\phi \geq 0.01143\pi$ for arbitrary $s$. Table 6 lists the first-order perturbation theory corrections away from the antiferromagnetic Ising limit $\omega = 0$ for arbitrary $s$, higher $S^z$ and three different values of $\phi$: one close to the

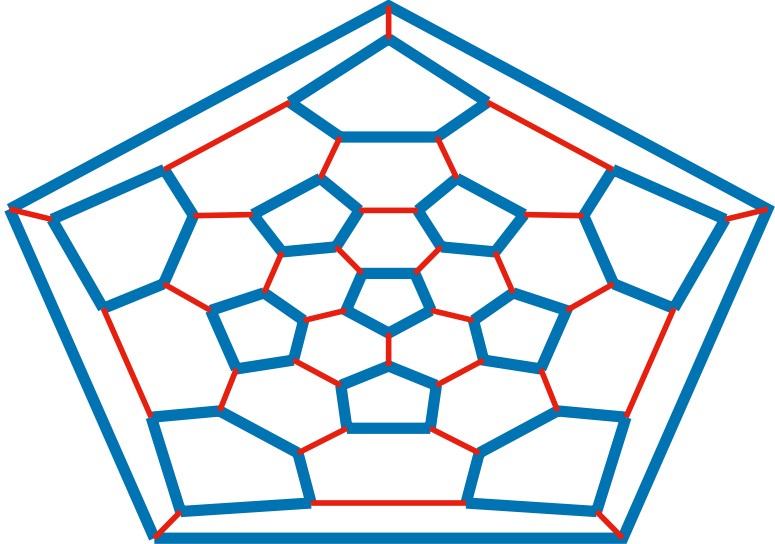

Figure 8: Planar projection of the truncated icosahedron $C_{60}$. The (blue) thick lines correspond to the twelve pentagons and the intrapentagon bonds and their exchange interaction strength $J_1 \equiv cos\phi$, while the (red) thin lines to the interpentagon (dimer) bonds and their exchange interaction strength $J_2 \equiv sin\phi$.

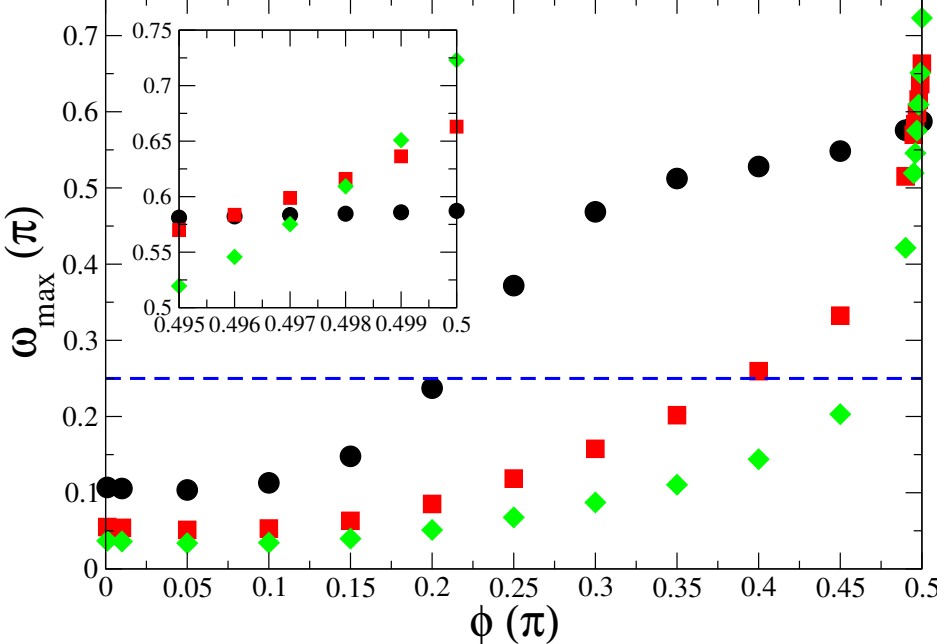

Figure 9: Maximum value $\omega_{max}$ for which the $s = \frac{1}{2}$ (black circles), 1 (red squares), and $\frac{3}{2}$ (green diamonds) ground state of Hamiltonian (1) for $C_{60}$ has a $\Delta S^z = 2$ magnetization discontinuity between $S^z = Ns-6$ and $Ns-4$ as a function of $\phi = tan^{-1}\frac{J_2}{J_1}$, which determines the relative strength of the two symmetrically independent exchange interactions. The dashed line shows the isotropic Heisenberg limit $\omega = \frac{\pi}{4}$. The inset focuses on the high range of $\phi$.

isolated pentagon limit $J_2 \ll J_1$ ($\phi = \frac{\pi}{50}$), one corresponding to uniform exchange interactions ($\phi = \frac{\pi}{4}$), and one close to the dimer limit $J_2 \gg J_1$ ($\phi = \frac{49}{100}\pi$). Like for the dodecahedron, the sector with five spin-flips from saturation is never the ground state in a magnetic field (Figs 4(b), (c), and (d)). The $\Delta S^z = 2$ discontinuity of the dodecahedron is inherited from

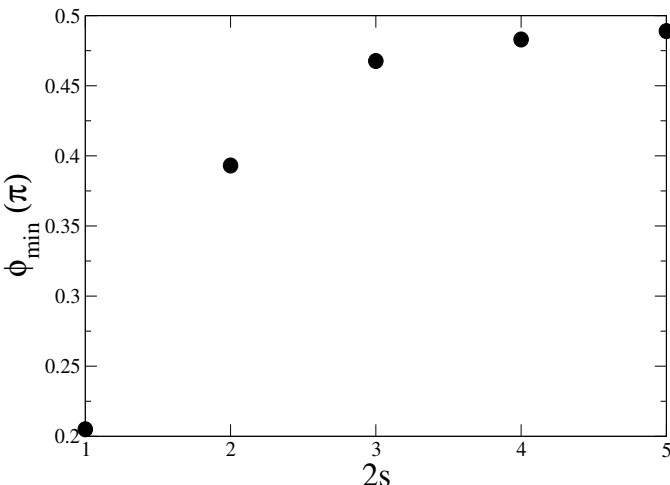

Figure 10: Minimum value $\phi_{min}$ as a function of $s$ for which the ground state of Hamiltonian (1) for $C_{60}$ has a $\Delta S^z = 2$ magnetization discontinuity five spin flips away from saturation at the isotropic Heisenberg limit $\omega = \frac{\pi}{4}$.

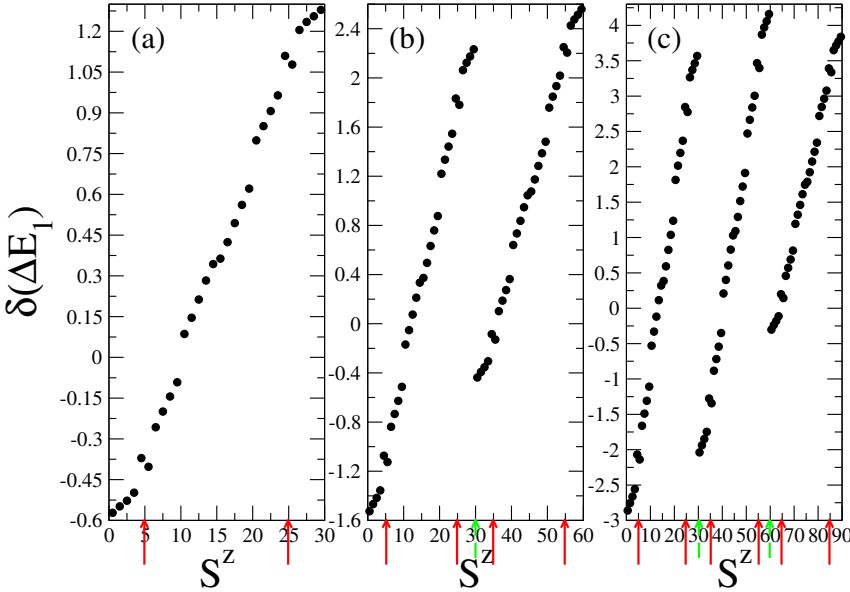

Figure 11: Differences of the first-order degenerate perturbation theory corrections of the energy $\delta(\Delta E_1) = \Delta E_1(S^z + 1) - \Delta E_1(S^z)$ between successive $S^z$ sectors of $C_{60}$ away from the dimer limit $\phi = \frac{\pi}{2}$ at the antiferromagnetic isotropic Heisenberg limit $\omega = \frac{\pi}{4}$ of Hamiltonian (1) for $h = 0$ and (a) $s = \frac{1}{2}$ ($\Delta E_1$ given in Table 10), (b) $s = 1$ ($\Delta E_1$ given in Table 11), and (c) $s = \frac{3}{2}$ ($\Delta E_1$ given in Table 12). The (red) solid arrows show the locations of the $\Delta S^z = 2$ magnetization discontinuities, and the (green) long-dashed arrows the locations of standard $\Delta S^z = 1$ discontinuities which are due to the unperturbed energies (App. A).

the truncated icosahedron, which shares its spatial symmetry, within first-order degenerate perturbation theory away from the Ising limit, as long as the interpentagon coupling is not very small. Further evidence to that is provided by the irreducible representations that include the lowest-energy state in each $S^z$ sector, which are very similar for the two molecules.

Table 6: Hamiltonian (1) with $h = 0$ for $C_{60}$ for arbitrary $s$ and three different values of $\phi = tan^{-1}\frac{J_2}{J_1}$ that determines the relative strength of the symmetrically independent exchange interactions. $\phi = \frac{\pi}{50}$ is close to the isolated pentagon limit $J_2 \ll J_1$, $\phi = \frac{\pi}{4}$ corresponds to uniform exchange interactions, and $\phi = \frac{49}{100}\pi$ is close to the dimer limit $J_2 \gg J_1$. The columns list the sector $S^z$ (with saturation value $Ns$), and for each $\phi$ value the first-order degenerate perturbation theory energy correction per spin magnitude $\frac{\Delta E_1}{s}$ on $\omega$ (away from the Ising limit $\omega = 0$), its degeneracy (deg.), and the irreducible representation (irrep.) of the $I_h$ symmetry group it belongs. $\Delta E_1$ has been calculated with double-precision accuracy but less digits are shown.

| $S^z$ | $\phi = \frac{\pi}{50}$ | | | $\phi = \frac{\pi}{4}$ | | | $\phi = \frac{49}{100}\pi$ | | |
| | $\frac{\Delta E_1}{s}$ | deg. | irrep. | $\frac{\Delta E_1}{s}$ | deg. | irrep. | $\frac{\Delta E_1}{s}$ | deg. | irrep. |
|---|---|---|---|---|---|---|---|---|---|
| $Ns-7$ | -11.41820 | 4 | $F_u$ | -11.43936 | 4 | $F_u$ | -7.19164 | 4 | $F_u$ |
| $Ns-6$ | -9.89498 | 1 | $A_g$ | -10.22177 | 1 | $A_g$ | -6.20833 | 1 | $A_g$ |
| $Ns-5$ | -8.28299 | 4 | $F_u$ | -8.69749 | 3 | $T_{1g}$ | -5.19086 | 4 | $F_u$ |
| $Ns-4$ | -6.67365 | 1 | $A_u$ | -7.19834 | 1 | $A_u$ | -4.17734 | 1 | $A_u$ |
| $Ns-3$ | -5.01822 | 3 | $T_{1g}$ | -5.46684 | 3 | $T_{1g}$ | -3.14149 | 3 | $T_{1g}$ |
| $Ns-2$ | -3.35042 | 5 | $H_g$ | -3.67399 | 5 | $H_g$ | -2.09766 | 5 | $H_g$ |
| $Ns-1$ | -1.67763 | 3 | $T_{2g}$ | -1.85123 | 3 | $T_{2g}$ | -1.05033 | 3 | $T_{2g}$ |
| $Ns$ | 0 | 1 | $A_g$ | 0 | 1 | $A_g$ | 0 | 1 | $A_g$ |

For the dodecahedron it was found that the fluctuations around the Ising axis work against the $\Delta S^z = 2$ ground-state magnetization discontinuity, and eventually a sufficiently strong value of $\omega$ makes the discontinuity disappear. The same is true for the truncated icosahedron, and Fig. 9 plots the maximum value $\omega_{max}$ for which the $\Delta S^z = 2$ magnetization discontinuity exists as a function of $\phi$ for $s$ up to $\frac{3}{2}$ (the corresponding data is listed in Table 7 and have been calculated with Lanczos diagonalization). For $\phi < 0.01143\pi$ it takes a minimum value $\omega_{min}$ for the jump to appear, and $\omega_{max}$ is small and slightly decreases with $\phi$. The $\omega_{min}$ values are listed in Table 8 for $s = \frac{1}{2}$, 1, and $\frac{3}{2}$. Within the accuracy of the calculation they are inversely proportional to $2s$. As $\phi$ increases the interpentagon bonds get stronger at the expense of the intrapentagon bonds, resulting in a discontinuity that survives up to higher values of $\omega$, until a sufficiently strong $\phi$ supports the jump up to the isotropic limit $\omega = \frac{\pi}{4}$.

The $\omega_{max}$ value for which the discontinuity survives decreases with $s$ for fixed $\phi$ according to Fig. 9, showing that the interactions in the $xy$ plane are becoming more detrimental to the jump with increasing $s$, as has also been found for the dodecahedron. Simultaneously, the required minimum value $\phi_{min}$ for the jump to survive at the isotropic Heisenberg limit increases with $s$, and is already close to $\frac{\pi}{2}$ for $s = \frac{3}{2}$ (Fig. 10 and Table 9). These results demonstrate that the stronger the interaction between the pentagons the more "dodecahedron-like" the truncated icosahedron becomes in terms of $\Delta S^z = 2$ discontinuous ground-state magnetic response. They also explain the common magnetic properties of the two molecules, and also point to a similar mechanism for bigger $I_h$ fullerenes that also share these properties [30].

The inset of Fig. 9 focuses on values of $\phi$ close to $\frac{\pi}{2}$. Contrary to what happens for smaller $\phi$, $\omega_{max}$ increases with $s$ as $\phi$ approaches $\frac{\pi}{2}$. When $\phi \to \frac{\pi}{2}$ the truncated icosahedron reduces to 30 dimers perturbatively coupled via intrapentagon bonds, which form an icosidodecahedron. First-order degenerate perturbation theory gives the lowest-energy correction $\Delta E_1$ at

Table 7: Maximum value $\omega_{max}$ for which the $s = \frac{1}{2}$, 1, and $\frac{3}{2}$ ground state of Hamiltonian (1) for $C_{60}$ has a $\Delta S^z = 2$ magnetization discontinuity between $S^z = Ns - 6$ and $Ns - 4$ for different values of $\phi = tan^{-1}\frac{J_2}{J_1}$, which determines the relative strength of the two symmetrically independent exchange interactions. The values for $0.5\pi^-$ are found from first-order perturbation theory on the dimer limit $\phi = \frac{\pi}{2}$ (Table 13).

| $\phi(\pi)$ | $\omega_{max}(s = \frac{1}{2})(\pi)$ | $\omega_{max}(s = 1)(\pi)$ | $\omega_{max}(s = \frac{3}{2})(\pi)$ |
|---|---|---|---|
| 0.001 | 0.10693 | 0.05497 | 0.03684 |
| 0.01 | 0.10558 | 0.05385 | 0.03598 |
| 0.05 | 0.10358 | 0.05114 | 0.03375 |
| 0.1 | 0.11293 | 0.05304 | 0.03439 |
| 0.15 | 0.14775 | 0.06311 | 0.03971 |
| 0.2 | 0.23722 | 0.08522 | 0.05109 |
| 0.25 | 0.37183 | 0.11856 | 0.06764 |
| 0.3 | 0.46865 | 0.15763 | 0.08718 |
| 0.35 | 0.51236 | 0.20175 | 0.11042 |
| 0.4 | 0.52813 | 0.25949 | 0.14385 |
| 0.45 | 0.54835 | 0.33235 | 0.20304 |
| 0.49 | 0.57578 | 0.51511 | 0.42140 |
| 0.495 | 0.58107 | 0.56995 | 0.51947 |
| 0.496 | 0.58223 | 0.58366 | 0.54571 |
| 0.497 | 0.58342 | 0.59883 | 0.57525 |
| 0.498 | 0.58464 | 0.61600 | 0.60929 |
| 0.499 | 0.58591 | 0.63632 | 0.65086 |
| $0.5^-$ | 0.58723 | 0.66303 | 0.72301 |

Table 8: Minimum value $\omega_{min}$ for which the $s = \frac{1}{2}$, 1, and $\frac{3}{2}$ ground state of Hamiltonian (1) for $C_{60}$ has a $\Delta S^z = 2$ magnetization discontinuity between $S^z = Ns - 6$ and $Ns - 4$ for different values of $\phi = tan^{-1}\frac{J_2}{J_1}$, which determines the relative strength of the two symmetrically independent exchange interactions. For $\phi \geq 0.01143\pi$ it is $\omega_{min} = 0$ for arbitrary $s$.

| $\phi(\pi)$ | $\omega_{min}(s = \frac{1}{2})(\pi)$ | $\omega_{min}(s = 1)(\pi)$ | $\omega_{min}(s = \frac{3}{2})(\pi)$ |
|---|---|---|---|
| 0.001 | 0.00681 | 0.00341 | 0.00227 |
| 0.01 | 0.00089 | 0.00044 | 0.00030 |

the isotropic Heisenberg limit $\omega = \frac{\pi}{4}$ for every $S^z$ sector, and not only close to saturation as with Lanczos diagonalization, as long as $s$ is not too big (Tables 10, 11, and 12). Figure 11 plots the differences $\delta(\Delta E_1)$ between successive $S^z$ sectors (App. A). These typically increase with $S^z$, resulting in magnetization jumps $\Delta S^z = 1$. For $S^z$ sectors which are multiples of 5 the dependence of $\delta(\Delta E_1)$ on $S^z$ is not as smooth, and in particular around the sectors $S^z = \frac{N}{2}i + 5$ and $S^z = \frac{N}{2}(i + 1) - 5$ with $i = 0, 1, \ldots, 2s - 1$ $\delta(\Delta E_1)$ decreases with $S^z$. These sectors never include the ground state in a magnetic field, resulting in a number of $4s$ discontinuities as a function of $s$ with $\Delta S^z = 2$. These are sectors with an $S^z$ value differing by 5 from sectors whose $S^z$ is an integer multiple of $\frac{N}{2}$. The appearance of $\Delta S^z = 2$ ground-state discontinuities in pairs for regions of $S^z$ where $\frac{N}{2}i \leq S^z \leq \frac{N}{2}(i + 1)$ originates from the hole-particle symmetry with respect to the center of the region, after the projection to the lowest-energy dimer states

Table 9: Minimum value $\phi_{min}$ as a function of $s$ for which the ground state of Hamiltonian (1) for $C_{60}$ has a $\Delta S^z = 2$ magnetization discontinuity between $S^z = Ns - 6$ and $Ns - 4$ at the isotropic Heisenberg limit $\omega = \frac{\pi}{4}$.

| $s$ | $\phi_{min}(\pi)$ |
|---|---|
| $\frac{1}{2}$ | 0.20503 |
| 1 | 0.39310 |
| $\frac{3}{2}$ | 0.46766 |
| 2 | 0.48299 |
| $\frac{5}{2}$ | 0.48894 |

has been made [44–46]. Since Lanczos diagonalization shows that the $\Delta S^z = 2$ discontinuity five spin flips away from saturation is present at the isotropic Heisenberg limit away from the perturbative dimer limit the further the smaller $s$ is, it is expected that the other discontinuities can also survive relatively far from the strong dimer limit, allowing to infer the magnetic response for not-so-high $S^z$, where the Hilbert space is enormous, from the dimer limit where the size of the Hilbert space is tractable. Table 13 lists the $\omega_{max}$ values for the different discontinuities when $\phi \to \frac{\pi}{2}$, which increase with $s$ as was also found in the inset of Fig. 9. Each value belongs to a pair of discontinuities.

Table 10: First-order degenerate perturbation theory correction of the energy $\Delta E_1$ for the different $S^z$ sectors of $C_{60}$ away from the dimer limit $\phi = \frac{\pi}{2}$ at the antiferromagnetic isotropic Heisenberg limit $\omega = \frac{\pi}{4}$ of Hamiltonian (1) for $h = 0$ and $s = \frac{1}{2}$.

| $S^z$ | $\Delta E_1$ | $S^z$ | $\Delta E_1$ | $S^z$ | $\Delta E_1$ | $S^z$ | $\Delta E_1$ |
|---|---|---|---|---|---|---|---|
| 0 | 0 | 8 | -3.37593 | 16 | -2.17645 | 24 | 3.44460 |
| 1 | $-\frac{\sqrt{2}}{8}(\sqrt{5}+1)$ | 9 | -3.51988 | 17 | -1.75230 | 25 | 4.55445 |
| 2 | -1.12036 | 10 | -3.61148 | 18 | -1.25823 | 26 | 5.63228 |
| 3 | -1.64789 | 11 | -3.52543 | 19 | -0.69700 | 27 | 6.83739 |
| 4 | -2.14589 | 12 | -3.37955 | 20 | -0.07595 | 28 | 8.07203 |
| 5 | -2.51661 | 13 | -3.16651 | 21 | 0.72276 | 29 | $\frac{\sqrt{2}}{8}(55-\sqrt{5})$ |
| 6 | -2.91936 | 14 | -2.88356 | 22 | 1.57382 | 30 | $\frac{15}{2}\sqrt{2}$ |
| 7 | -3.17653 | 15 | -2.54002 | 23 | 2.48032 | | |

Table 11: Same as Tab. 10 for $s = 1$.

| $S^z$ | $\Delta E_1$ | $S^z$ | $\Delta E_1$ | $S^z$ | $\Delta E_1$ | $S^z$ | $\Delta E_1$ |
|---|---|---|---|---|---|---|---|
| 0 | 0 | 16 | -9.90661 | 31 | 10.16959 | 46 | 15.11256 |
| 1 | $-\frac{\sqrt{2}}{3}(\sqrt{5}+1)$ | 17 | -9.41163 | 32 | 9.77693 | 47 | 16.28654 |
| 2 | -2.99310 | 18 | -8.77918 | 33 | 9.42314 | 48 | 17.57092 |
| 3 | -4.40984 | 19 | -8.01996 | 34 | 9.11786 | 49 | 18.95782 |
| 4 | -5.76557 | 20 | -7.14364 | 35 | 9.03347 | 50 | 20.43860 |
| 5 | -6.83962 | 21 | -5.92341 | 36 | 8.90460 | 51 | 22.19653 |
| 6 | -7.96525 | 22 | -4.58861 | 37 | 9.00707 | 52 | 24.04421 |
| 7 | -8.80401 | 23 | -3.14716 | 38 | 9.19497 | 53 | 25.97763 |
| 8 | -9.53836 | 24 | -1.60129 | 39 | 9.46861 | 54 | 27.99649 |
| 9 | -10.16605 | 25 | 0.23145 | 40 | 9.83200 | 55 | 30.24667 |
| 10 | -10.67917 | 26 | 2.01260 | 41 | 10.47254 | 56 | 32.45238 |
| 11 | -10.84838 | 27 | 4.07544 | 42 | 11.20696 | 57 | 34.87899 |
| 12 | -10.90051 | 28 | 6.19929 | 43 | 12.04390 | 58 | 37.35409 |
| 13 | -10.82584 | 29 | $\frac{\sqrt{2}}{3}(20-\sqrt{5})$ | 44 | 12.99124 | 59 | 39.86807 |
| 14 | -10.61371 | 30 | $\frac{15}{2}\sqrt{2}$ | 45 | 14.03629 | 60 | $30\sqrt{2}$ |
| 15 | -10.27938 | | | | | | |

Table 12: Same as Tab. 10 for $s = \frac{3}{2}$.

| $S^z$ | $\Delta E_1$ | $S^z$ | $\Delta E_1$ | $S^z$ | $\Delta E_1$ | $S^z$ | $\Delta E_1$ |
|---|---|---|---|---|---|---|---|
| 0 | 0 | 23 | -11.04769 | 46 | 2.08329 | 69 | 43.65523 |
| 1 | $-\frac{5\sqrt{2}}{8}(\sqrt{5}+1)$ | 24 | -8.67899 | 47 | 3.37494 | 70 | 44.47147 |
| 2 | -5.61564 | 25 | -5.83354 | 48 | 4.89081 | 71 | 45.66505 |
| 3 | -8.27830 | 26 | -3.05808 | 49 | 6.61187 | 72 | 46.98738 |
| 4 | -10.83625 | 27 | 0.20699 | 50 | 8.52480 | 73 | 48.44870 |
| 5 | -12.90461 | 28 | 3.57675 | 51 | 10.99590 | 74 | 50.06102 |
| 6 | -15.04295 | 29 | $\frac{\sqrt{2}}{8}(51-5\sqrt{5})$ | 52 | 13.66065 | 75 | 51.80777 |
| 7 | -16.70454 | 30 | $\frac{15}{2}\sqrt{2}$ | 53 | 16.49980 | 76 | 53.59655 |
| 8 | -18.19429 | 31 | 8.56781 | 54 | 19.50498 | 77 | 55.51977 |
| 9 | -19.50325 | 32 | 6.62997 | 55 | 22.97068 | 78 | 57.59398 |
| 10 | -20.61117 | 33 | 4.78123 | 56 | 26.36796 | 79 | 59.80719 |
| 11 | -21.14082 | 34 | 3.03344 | 57 | 30.23707 | 80 | 62.14914 |
| 12 | -21.47156 | 35 | 1.75747 | 58 | 34.20713 | 81 | 64.86843 |
| 13 | -21.58974 | 36 | 0.41310 | 59 | 38.26630 | 82 | 67.71467 |
| 14 | -21.47556 | 37 | -0.47076 | 60 | $30\sqrt{2}$ | 83 | 70.67810 |
| 15 | -21.15379 | 38 | -1.18859 | 61 | 42.12444 | 84 | 73.75551 |
| 16 | -20.76846 | 39 | -1.73202 | 62 | 41.88711 | 85 | 77.14699 |
| 17 | -20.17553 | 40 | -2.08180 | 63 | 41.70662 | 86 | 80.48426 |
| 18 | -19.35024 | 41 | -1.87341 | 64 | 41.59339 | 87 | 84.13303 |
| 19 | -18.31239 | 42 | -1.47315 | 65 | 41.79165 | 88 | 87.84905 |
| 20 | -17.07563 | 43 | -0.86770 | 66 | 41.93570 | 89 | 91.62191 |
| 21 | -15.26061 | 44 | -0.03803 | 67 | 42.39383 | 90 | $\frac{135}{2}\sqrt{2}$ |
| 22 | -13.24454 | 45 | 0.99192 | 68 | 42.96594 | | |

Table 13: Maximum value $\omega_{max}$ for which the $s = \frac{1}{2}$, 1, and $\frac{3}{2}$ first-order degenerate perturbation theory ground state of Hamiltonian (1) for $C_{60}$ away from the dimer limit $\phi = \frac{\pi}{2}$ has $\Delta S^z = 2$ magnetization discontinuities, where the sectors $S^z = \frac{N}{2}i + 5$ and $\frac{N}{2}(i+1) - 5$ with $i = 0, 1, \ldots, 2s - 1$ never include the ground state in the external field $h$.

| $s$ | $i$ | $\omega_{max}(\pi)$ |
|---|---|---|
| $\frac{1}{2}$ | 0 | 0.58723 |
| 1 | 0 | 0.72291 |
| 1 | 1 | 0.66303 |
| $\frac{3}{2}$ | 0 | 0.75- |
| $\frac{3}{2}$ | 1 | 0.75- |
| $\frac{3}{2}$ | 2 | 0.72301 |

## 5 Conclusions

In this paper it was shown that the $\Delta S^z = 2$ quantum magnetization discontinuity of the isotropic quantum antiferromagnetic Heisenberg model on $C_{20}$ in an external field can be continuously traced to a strong discontinuity present at the Ising limit of the model for finite $s$. This discontinuity originates from the special connectivity of the dodecahedron. $C_{60}$, which also has $I_h$ symmetry, inherits the discontinuity also at the isotropic limit for sufficiently strong interpentagon interactions. Perturbing away from the limit of infinitely strong interpentagon interactions the number of $\Delta S^z = 2$ discontinuities equals $4s$, and these discontinuities are expected to survive at least close to this limit.

Since the $N = 80$ $I_h$-symmetry fullerene, the chamfered dodecahedron [1], also has a jump in a magnetic field for $s = \frac{1}{2}$ when its two symmetrically unique interactions are equal [30], it is expected that bigger molecules with the same symmetry will follow the $\Delta S^z = 2$ discontinuity pattern of $C_{60}$. In their case the interpentagon coupling depends on the strength of exchange interactions of one or more kinds.

The results of the paper show how the magnetization response is determined by the spatial symmetry for the $I_h$ fullerene molecules. The twelve pentagons are distributed according to a specific symmetry that is directly connected with how the spins react to an external magnetic field. This can lead to the prediction of the magnetic properties of large molecules, which are computationally intractable, through the treatment of their smaller symmetric relatives. Especially at the strong-coupling limit of the twelve pentagons perturbation theory can generate the magnetic response for the whole $S^z$ range, and not only for the higher $S^z$ numbers, which are accessible with Lanczos diagonalization. The response at the dimer limit provides insights into the magnetization response at least close to this limit, as has been shown for the $\Delta S^z = 2$ discontinuity close to saturation. It is highly desirable to find correlations between magnetic behavior and spatial symmetry [11, 27, 29, 30, 51, 52]. In addition, and since the AHM is the large-$U$ limit of the Hubbard model at half-filling [17, 18], it is expected that the $I_h$-symmetry fullerene molecules will share properties related to strongly-correlated models of itinerant electrons, at least for some range of their parameters.

# A Degenerate Perturbation Theory on the Intrapentagon Coupling for $C_{60}$

As an example, the case $s = 1$ is considered. The interaction between nearest-neighbor spins belonging to different pentagons is given by the term inside the brackets in Hamiltonian (1) and $0 \leq \omega < \frac{3}{4}\pi$ is taken. The lowest energies of these dimers in the different $S^z_{dimer}$ sectors are

$$
\begin{aligned}
S^z_{dimer} &= 0, \qquad e_0 = -\frac{1}{2}\left(\cos\omega + \sqrt{1 + 7\sin^2\omega}\right), \\
S^z_{dimer} &= 1, \qquad e_1 = -\sin\omega, \\
S^z_{dimer} &= 2, \qquad e_2 = \cos\omega.
\end{aligned}
\tag{A.1}
$$

It is $e_0 < e_1 < e_2$ for $0 \leq \omega < \frac{3}{4}\pi$. The corresponding dimer eigenstates are called $|0>$, $|1>$, and $|2>$ respectively. The unperturbed Hamiltonian (1) at the $\phi = \frac{\pi}{2}$-limit for $C_{60}$ has only the interpentagon bonds different from zero, resulting in isolated dimers. The lowest-energy unperturbed states for the different $S^z$ sectors are given as sets of the number of dimers found in each one of the three different dimer eigenstates (# in $|0>$, # in $|1>$, # in $|2>$) = $(n_0, n_1, n_2)$ (Table 14). The corresponding unperturbed energies are $E_0(S^z) = n_0 e_0 + n_1 e_1 + n_2 e_2$. Interdimer interactions, which are nearest-neighbor intrapentagon interactions, are taken as a perturbation to the $\phi = \frac{\pi}{2}$-limit of Hamiltonian (1). Tables 10, 11, and 12 list the first-order degenerate perturbation theory energy corrections $\Delta E_1$ of the intrapentagon bonds on the $\phi = \frac{\pi}{2}$-dimer limit of Hamiltonian (1) for $s = \frac{1}{2}$, 1, and $\frac{3}{2}$ at the isotropic Heisenberg limit $\omega = \frac{\pi}{4}$.

Within first-order perturbation theory the lowest-energy levels in the different $S^z$ sectors are given as $E_0(S^z) + \Delta E_1(S^z)\lambda$, with $\lambda \equiv \frac{J_1}{J_2}$ the perturbation parameter away from the dimer limit. The difference between successive unperturbed energies $E_0(S^z + 1) - E_0(S^z)$ is the same for $\frac{N}{2}i \leq S^z \leq \frac{N}{2}(i+1) - 1$, $i = 0, 1, \dots, 2s - 1$. For $s = 1$ there are two such $S^z$ ranges, one between $S^z = 0$ to 30 and the second between $S^z = 30$ and 60 (Fig. 11(b)). At the borders between these $2s$ regions the magnetization step $\Delta S^z = 1$, due to the unperturbed energies. Within first-order perturbation theory the magnitude of $\Delta S^z$ within the $2s$ regions is determined by the difference in the perturbative energy corrections between successive sectors $[\Delta E_1(S^z + 1) - \Delta E_1(S^z)]\lambda = \delta(\Delta E_1)\lambda$, which is taken to correspond to the middle point $S^z + \frac{1}{2}$. These differences are symmetric with respect to the center of each region, with symmetrically placed differences adding up to $\frac{2i+1}{\sqrt{2}}$. This is a manifestation of the hole-particle symmetry with respect to the center of the region, after the projection to the lowest-energy dimer states has been made. If these differences do not increase with $S^z$ the magnetization step $\Delta S^z > 1$. According to Tables 10, 11, and 12 and Fig. 11 there are $4s$ discontinuities for a specific $s$, with the sectors $S^z = \frac{N}{2}i + 5$ and $S^z = \frac{N}{2}(i+1) - 5$ with $i = 0, 1, \dots, 2s - 1$ never including the ground state in a magnetic field. These are sectors whose $S^z$ value differs by 5 from the sectors $S^z = \frac{N}{2}i$, with $i = 0, 1, \dots, 2s$. Table 13 lists the $\omega_{max}$ values for the $4s$ discontinuities for every $s$, with each value belonging to a pair of discontinuities.

Table 14: Lowest-energy unperturbed states for Hamiltonian (1) for the truncated icosahedron for $\phi = \frac{\pi}{2}$ and $s = 1$ as a function of $S^z$ given as sets of the number of dimers found in each one of the three different dimer eigenstates (# in $|0>$,# in $|1>$,# in $|2>$) = $(n_0, n_1, n_2)$, and their number.

| $S^z$ | dimer eigenstate types | number of states |
|---|---|---|
| 60 | (0,0,30) | 1 |
| 59 | (0,1,29) | 30 |
| 58 | (0,2,28) | 435 |
| 57 | (0,3,27) | 4060 |
| 56 | (0,4,26) | 27405 |
| 55 | (0,5,25) | 142506 |
| 54 | (0,6,24) | 593775 |
| ⋮ | ⋮ | ⋮ |
| 32 | (0,28,2) | 435 |
| 31 | (0,29,1) | 30 |
| 30 | (0,30,0) | 1 |
| 29 | (1,29,0) | 30 |
| 28 | (2,28,0) | 435 |
| 27 | (3,27,0) | 4060 |
| 26 | (4,26,0) | 27405 |
| 25 | (5,25,0) | 142506 |
| 24 | (6,24,0) | 593775 |
| ⋮ | ⋮ | ⋮ |
| 2 | (28,2,0) | 435 |
| 1 | (29,1,0) | 30 |
| 0 | (30,0,0) | 1 |

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
