# Peer review of "$\Delta S^z=2$ quantum magnetization discontinuities proportional in number to the spin $s$ in C$_{60}$: origin and the role of symmetry"

_SciPost Physics, doi:SciPost Phys. 15, 037 (2023)_

## Round 1 · Referee Report · Anonymous · 2023-1-3

Strengths
1. Gets at the salient properties of magnetization curves for large molecules using insights into the problem rather than hard numerics
Weaknesses
1. The results are not entirely original, but extend KarǏová et al. (2017) [43] and Konstantinidis (2007) [30]
2. The presentation can be further improved
Report
The paper deals with the magnetization jumps of the quantum XXZ model on the C20 and C60 geometries and attempts to trace these discontinuities between the Ising, Heisenberg and XX limits. I think that the results are quite interesting, but the presentation can be significantly improved by pointing out the insights more explicitly.
Requested changes
I suggest that the author improves the presentation by considering the points below:
1. The high-field discontinuity of the Heisenberg case is continuously connected to the case of weak XX coupling, which is more easily tractable using perturbation theory and can be better understood conceptually. This simplification aspect should be stressed more.
2. The high-field discontinuity around Sz=25 for C60 seems to be connected to a low-field discontinuity around Sz=5 via the particle-hole symmetry. This is at least the case for weak interpentagon coupling and a case can be made that is survives up to the Heisenberg limit. I would stress this point much more strongly, since it allows to infer the properties of a region with a very large Hilbert space from those of a region with a modest Hilbert space.
3. There seems to be a kind of universal behavior of the I_h molecules close to saturation, with the same irrep sequence and a high-spin discontinuity (also a corresponding low-spin discontinuity?). I would say this is not trivial and should be stressed more, as molecules are generally individualistic and non-universal.
Some minor points that are unclear or require changes/corrections are:
1. It would be good to explicitly show some curves of M(h), perhaps as side panels for selected values of omega. I think it will be otherwise difficult for non-specialists to understand the results shown in Figs. 1, 2.
2. E_1 should probably be changed to \Delta E_1, as it is a perturbative correction and not the absolute value of the energy.
3. The introduction mentions both classical and quantum spins and it is sometimes confusing which one is meant. I would suggest using AQHM instead of AHM and write "quantum spins" whenever possible and applicable.
4. The word "controlled" on p. 7 should probably be deleted; similar with "can be tuned" in the abstract. The interpentagon interactions cannot be controlled experimentally. I would regard tuning them as a theoretical device.
5. Some typos: p.2: "originates in"->"originates from"? p.7: "effected"->"affected"
6. For C60, using Lanczos diagonalization to get the exact energies for 5-6 spinflips below saturation should be doable. Is this done? And if not - why not?
7. On p.6, it is stated that "quantum fluctuations are detrimental to the jump", but I would interpret the results of Fig. 3 in an opposite way: Increasing s for a fixed omega suppresses quantum fluctuations and removes the jump. The problem seems to be that there are two ways of defining a classical limit: (a) omega→0 (Ising case) or (b) s→∞. Some more clarity should be brought into this.
8. The final sentence "such common magnetic behavior [...] is well sought after" is somewhat cryptic and should be elaborated.
Author: Nikolaos Konstantinidis on 2023-04-03 [id 3532]
(in reply to Report 1 on 2023-01-03)
Referee 1
I thank Referee 1 for the report. The changes in the text are highlighted in red.
For the truncated icosahedron I have added figure 10 and tables 8 and 9, to plot the minimum value of \phi for which there is a \Delta S^z = 2 discontinuity five spin flips away from saturation at the isotropic Heisenberg limit as a function of s, and the minimum value of \omega for which a discontinuity appears five spin flips from saturation for very small \phi as a function of s.
Weaknesses
-
The topic of KarǏová et al. (2017) [43] is not related to the current paper. Red text has been added 15 lines before the end of page 2 explaining that paper [43] is extending the calculation done in paper [29] by introducing an anisotropy parameter, but no detailed connection with the Ising limit is attempted as the paper has a different topic. In order to establish the presence of the Ising discontinuity for any value of the interactions in the xy plane degenerate perturbation theory is required, which is introduced in the current paper. Furthermore in [43] there is no discussion whatsoever of the truncated icosahedron, the similarities of its magnetic properties with the dodecahedron with which it shares the spatial symmetry, or the presence of multiple discontinuities close to the dimer limit and the particle-hole symmetry, which form the bulk of the current article. In the paper of Konstantinidis (2007) [30] again the origin of the discontinuity is not discussed. Even though it is shown that the I_h clusters share the high-field discontinuity, there is no explanation why or any hint of the multiple discontinuities of the truncated icosahedron close to the dimer limit and the particle-hole symmetry.
-
The paper has been expanded with the addition of new text and figures and the presentation has been improved.
Requested changes
-
A sentence has been added in the abstract, two sentences before its end. More text has been added on: -the second page 13 lines before its end. -page 3 line 6, where it is stated that for the truncated icosahedron the discontinuity appears already at the Ising limit provided \phi exceeds a (small) minimum value. -page 4, beginning of the second paragraph of section 3: "Perturbing away from the Ising limit". -page 6, beginning of the second paragraph of section 4: again it is stated that for the truncated icosahedron the discontinuity appears already at the Ising limit provided \phi exceeds a (small) minimum value. More text has been added toward the end of the same paragraph.
-
Text has been added at the following points: -page 2, line 13 of the second paragraph: it has been added that it is not possible to calculate the magnetization response for the whole S^z-range for C_{60}, stating the problem. -page 3, 4th line of the last paragraph: again the problem with the size of the Hilbert space is referred to, and how degenerate perturbation theory allows insight very close to the dimer limit. The second sentence before the end of the paragraph adds more text. -page 7, line 5 of the last paragraph states that the magnetization response has been found for the whole S^z range. -page 8, end of section 4. -page 9, second and third sentences in red text in the last paragraph.
-
Text has been added at the following points: -"and the role of symmetry" has been added in the abstract. -page 3: text has been added at the end of the first paragraph on the symmetry patterns detected for the magnetic response of the I_h molecules. -page 9, first sentence in red text in the last paragraph.
Minor points
- Figures 5 and 6 have been added.
- The symbol has been changed to \Delta E_1.
- I have kept the notation AHM because sometimes classical spins are referred to, and have added "quantum" before AHM at appropriate points in the text with red color.
-
Indeed the tuning of the interactions was meant as a theoretical device. "Controlled" etc. have been removed and replaced with appropriate wording at the different points in the text, removing any implication that the interactions can be experimentally regulated.
-
The first word has been changed, the second is not part of the text any more.
-
Yes this has been done. The text now has more information at the points in the text were the calculations are mentioned about the method with which they were generated. A couple of clarifying statements have also been added in red text in section 2 (Model). Figures 2, 3, 5, 6, 7, 9, and 10 have been generated with Lanczos diagonalization. It is possible to find the lowest-energy level for any S^z sector with Lanczos diagonalization for C_{20} for s=1/2 and 1, while for C_{60} Lanczos diagonalization can produce the lowest-energy states of the S^z sectors that include the five-spin-flips from saturation discontinuity for any s. Typically captions of figures and tables specifically mention if the results have been generated with degenerate perturbation theory. On the other hand Lanczos diagonalization is not explicitly mentioned, and corresponds to calculations five spin flips away from saturation.
-
The text has been corrected to "the interactions in the $xy$ plane are becoming more detrimental to the jump with increasing $s$". The sentence is now at the beginning of the second paragraph on page 7. What happens is exactly the same with what happens with the dodecahedron.
-
Text in red has been added in the last paragraph of section 5 that points to the importance of common magnetic behavior of symmetry-sharing molecules, as the properties of even larger molecules of the same symmetry that are computationally intractable can be inferred from the common patterns of their smaller symmetric relatives.
Author: Nikolaos Konstantinidis on 2023-04-03 [id 3534]
(in reply to Report 3 on 2023-01-05)I thank Referee 3 for the report. The changes in the text are highlighted in red.
For the truncated icosahedron I have added figure 10 and tables 8 and 9, to plot the minimum value of \phi for which there is a \Delta S^z = 2 discontinuity five spin flips away from saturation at the isotropic Heisenberg limit as a function of s, and the minimum value of \omega for which a discontinuity appears five spin flips from saturation for very small \phi as a function of s.
Weaknesses
-Figure 1 shows the dodecahedron.
-Figure 4 plots the differences of the first-order degenerate perturbation theory corrections over s away from the Ising limit for the dodecahedron and three different values of \phi for the truncated icosahedron.
-Figures 5 and 6 show the magnetization for the dodecahedron for three different values of \omega and s=1/2 and 1.
-Figure 8 shows the truncated icosahedron.
-Figure 10 plots the minimum value of \phi for which there is a $\Delta S^z=2$ magnetization discontinuity five spin flips away from saturation at the isotropic Heisenberg limit as a function of s.
-Figure 11 plots the differences of the first-order degenerate perturbation theory corrections away from the dimer limit at the isotropic Heisenberg limit for the truncated icosahedron and three different values of s. Arrows point at the S^z-locations of the magnetization discontinuities. This plots the information included in the corresponding three tables.
Requested changes
Figure 1 has been added and shows the dodecahedron, and figure 8 has been added and shows the truncated icosahedron. The latter figure has different colors distinguishing the two symetrically different types of edges.
In the beginning of section 3 (fourth line) it has been added with red letters that all J_{ij} = 1 for the dodecahedron, referring to Hamiltonian (1). In the beginning of section 4 (sixth line) the two symetrically unique exchange interactins have been explicitly written as J_1 \equiv cos\phi and J_2 \equiv sin\phi, also making reference to figure 8 that distinguishes the two with different colors. I believe that writing the Hamiltonian once for each molecule will be repetitive, since the differences between the two molecules are related to the exchange interactions, which are also visible in figure 8 for the truncated icosahedron, while the dot products of neighboring spins remain the same. Please comment if the Hamltonians are still not perfectly clear.
Toward the end of section 2, with text that is highlighted in red, it is added that the lowest-lying levels in the different S^z sectors are found with Lanczos diagonalization by taking the symmetries into account. This method has already been used in the past and a reference is given. Then it is added that at the Ising limit there are degenerate ground states in each S^z sector, necessitating the use of degenerate perturbation theory in order to find the first-order corrections, where symmetries are used as well. References to degenerate perturbation theory have also been added.
It has been added with red text in appropriate points in the manuscript if the results have been generated with Lanczos diagonalization or degenerate perturbation theory. Please comment if more information is required.
Typically captions of figures and tables specifically mention if the results have been generated with degenerate perturbation theory. On the other hand Lanczos diagonalization is not explicitly mentioned, and corresponds to calculations five spin flips away from saturation.
(the correct number here is 4) Figures 5 and 6 show the magnetization for the dodecahedron for three different values of \omega and s=1/2 and 1. The three different values of omega correspond to the limits suggested by the Referee.
(the correct number here is 5) Table 6 is discussed toward the bottom of page 6. The corresponding limits have been described (red text), and they are the isolated pentagon limit $J_2 \ll J_1$, the uniform exchange interactions case $J_2=J_1$, and the dimer limit $J_2 \gg J_1$.
(the correct number here is 6) The table is now Table 14 on page 24. It has a caption and does not exceed the limits of the page.

---

## Round 1 · Referee Report · Anonymous · 2023-1-4

Strengths
1. Smart approach for obtaining insights into magnetization properties of large molecules.
Weaknesses
1. Suboptimal slightly confusing presentation of the results.
2. The title of the paper can be improved.
Report
The paper addresses the magnetization discontinuities of the XXZ Heisenberg model on the C20 and C60 geometry. The approach is to trace the discontinuities close to the Ising limit towards the nontrivial parameter regime with finite XX interactions. In the C60 case, results are also obtained through tracing the discontinuities from the decoupled dimer limit towards the nontrivial isotropic C60 geometry.
The results are generally interesting and relevant but I think their presentation has to be improved (see requested changes).
I further have the feeling that the title of the paper does not reflect its whole content but only a part of it (see requested changes).
Requested changes
1. Add magnetization curves M(B) and also energy curves E(Sz). For example tables 8, 9 and 10 should be better visualized. In the current manuscript, it is very hard to understand the locations of the 4s discontinuities and also the effective particle-hole symmetry. After better visualization, some of the table data may be placed in an external link which provides all the numbers. This could improve readability.
2. Be more clear about the two different perturbation theories (in $\omega$ and in $\phi$). For example in the last paragraph of page 7 or in the caption of Table 8.
3. Rethink the title of the manuscript so that it is broader and not only mention the number of magnetization discontinuities of C60.
Author: Nikolaos Konstantinidis on 2023-04-03 [id 3533]
(in reply to Report 2 on 2023-01-04)
Referee 2
I thank Referee 2 for the report. The changes in the text are highlighted in red.
For the truncated icosahedron I have added figure 10 and tables 8 and 9, to plot the minimum value of \phi for which there is a \Delta S^z = 2 discontinuity five spin flips away from saturation at the isotropic Heisenberg limit as a function of s, and the minimum value of \omega for which a discontinuity appears five spin flips from saturation for very small \phi as a function of s.
Weaknesses
-
The paper has been expanded with the addition of new text and figures and the presentation has been improved.
-
"origin and the role of symmetry" was added at the end of the title to state that the paper elaborates on the origin of the discontinuities and points out the correlation between spatial symmetry and magnetic properties of the I_h fullerenes. The $\Delta S^z=2$ has been added at the beginning of the title to emphasize the discontinuities that are not common and are found here.
Requested changes
- Graphs have been added to visualize results.
-Figure 4 plots the differences of the first-order degenerate perturbation theory corrections over s away from the Ising limit for the dodecahedron and three different values of \phi for the truncated icosahedron.
-Figures 5 and 6 show the magnetization for the dodecahedron for three different values of \omega and s=1/2 and 1.
-Figure 11 plots the differences of the first-order degenerate perturbation theory corrections away from the dimer limit at the isotropic Heisenberg limit for the truncated icosahedron and three different values of s. Arrows point at the S^z-locations of the magnetization discontinuities. This figure plots the information included in the corresponding three tables.
-
(this is 2 in the correct numbering) Text has been added in the text and the captions of figures and tables to explain what the \omega and \phi values are in the different circumstances.
-
(this is 3 in the correct numbering) "origin and the role of symmetry" was added at the end of the title to state that the paper elaborates on the origin of the discontinuities and points out the correlation between spatial symmetry and magnetic properties of the I_h fullerenes. The $\Delta S^z=2$ has been added at the beginning of the title to emphasize the discontinuities that are not common and are found here.

---

## Round 1 · Referee Report · Anonymous · 2023-1-5

Strengths
1- Reports intriguing magnetization discontinuities in XXZ quantum Heisenberg spin clusters.
Weaknesses
1- Insufficient graphical presentation
2- Less clear explanation
Report
The author has studied XXZ AF Heisenberg spin clusters with the geometry of dodecahedron and fullerene by using diagonalization method and perturbation around Ising limit. The most interesting result is the theoretical prediction of magnetization discontinuity 4s which disappears in classical limit. The manuscript needs improvements in order to be publishable in highly reputable journal such as SciPost Physics.
Requested changes
1- The manuscript should be supplemented by graphical represantation of studied C20 and C60 molecules including the notation for two considered coupling constants in the latter molecule.
2- For better clarity, the manuscript should be supplemented by the explicit form of the Hamiltonians for the Heisenberg spin clusters with the geometry of C20 and C60 molecules. The former can be replace the general formula given by Eq. (1) and the latter should be added at the beginning of Sec. 4.
3-A few details of the calculation methods should be directly mentioned in the main manuscript and not only in Appendix where the technical details should be retained. It would be also valuable to explicitly mention in the manuscript which results were obtained from the diagonalization method and which come from perturbation calculations.
4- It would be beneficial if the manuscript would be supplemented by figures involving typical magnetization curves showing at least three possible scenarios with two magnetization discontinuities (easy axis case close to Ising limit), the isotropic and XX limiting case with only a single magnetization discontinuity.
5- The manuscript should contain at least brief note that the particular case \phi=\pi/100 is close to Ising limit and \phi=49\pi/100 is close to the XX limit auround the discussion of results of table 6.
6- The table on page 10 is without any caption and it flows over the page size.

---

## Round 2 · Referee Report · Anonymous · 2023-4-6

Report
The author has addressed all of my raised points and I can recommend the article for publication.
The journal's criteria are met because the complexity of the problem (C60 is roughly equivalent to a finite, but large periodic 2D system) is solved using insights rather than hard numerics. The ideas presented in the paper can be expanded to cases of even larger molecules or fermionic systems, where state-of-the-art numerical methods fail.

---

## Round 2 · Referee Report · Anonymous · 2023-4-28

Report
The author has added additional figures with the aim to clarify and facilitate the reading of the manuscript.
My general recommendation is still the publication in Scipost physics because of the insights into the magnetization plateaus of a large molecule with a smart approach (which can also be used for other molecules).
However, I still have concerns about the presentation. The additional figures help to clear the situation but due to the high amount of tables the reading flow is limited. In my opinion, it is worth to rethink if some of these data can be placed outside of the paper.
Requested changes
1) Fix typo in caption of Figure 11: remove parenthesis from `red` and `green`.
2) The graphical representation of C20 and C60 seems very large and the three-dimensionality is hard to sea. This should be improved or alternatively some projection to the 2D plane (e.g. Schlegel projection) should be used.
Report
The author has satisfactorily revised his manuscript in agreement with all suggestions and recommendations from my previous report. Besides, there has been made a lot of additional changes, which have substantionally improved quality and clarity of the manuscript. Bearing all this in mind, I recommend to accept the present version of the manuscript to SciPost Physics journal.

---

## Editorial Decision

published